# Development of an integrated and decentralised skin health strategy to improve experiences of skin neglected tropical diseases and other skin conditions in Atwima Mponua District, Ghana

**Richard Odame Phillips**[1,2]*, **Lucy Owusu**[1], **Eric Koka**[3], **Edmond Kwaku Ocloo**[3,4], **Hope Simpson**[5], **Abigail Agbanyo**[1], **Daniel Okyere**[4], **Ruth Dede Tuwor**[1], **Adelaide Fokuoh-Boadu**[1], **Richard Adjei Akuffo**[4], **Jacob Novignon**[1,6], **Michael Ntiamoah Oppong**[1], **Iris Mosweu**[7], **Adwoa Asante-Poku**[4], **Jojo Cobbinah**[8], **Tara B. Mtuy**[9], **Jennifer Palmer**[9], **Collins Ahorlu**[4], **Yaw Ampem Amoako**[1,2], **Stephen L. Walker**[9], **Dorothy Yeboah-Manu**[4], **Michael Marks**[5,9,10‡], **Catherine Pitt**[5‡], **Rachel Pullan**[5‡], **SHARP collaboration**[11]

**1** Kumasi Centre for Collaborative Research, Kwame Nkrumah University of Science & Technology, Kumasi, Ghana, **2** School of Medicine and Dentistry, Kwame Nkrumah University of Science & Technology, Kumasi, Ghana, **3** Department of Sociology and Anthropology, University of Cape Coast, Cape Coast, Ghana, **4** Noguchi Memorial Institute for Medical Research, University of Ghana, Accra, Ghana, **5** Faculty of Infectious and Tropical Diseases, London School of Hygiene & Tropical Medicine, London, United Kingdom, **6** Department of Economics, Kwame Nkrumah University of Science and Technology, Kumasi, Ghana, **7** Faculty of Public Health and Policy, London School of Hygiene & Tropical Medicine, London, United Kingdom, **8** Atwima Mponua District Health Directorate, Nhinahin, Ghana, **9** Hospital for Tropical Diseases, University College London Hospital, London, United Kingdom, **10** Division of Infection and Immunity, University College London, London, United Kingdom, **11** London School of Hygiene & Tropical Medicine, London, United Kingdom

‡ MM, CP and RP are joint senior authors on this work.
* rodamephillips@gmail.com

**Data Availability Statement:** All relevant data have been included in the manuscript.

## Abstract

Integrated strategies are recommended to tackle neglected tropical diseases of the skin (skin NTDs), which pose a substantial health and economic burden in many countries, including Ghana. We describe the development of an integrated and decentralised skin health strategy designed to improve experiences of skin NTDs in Atwima Mponua district in Ashanti Region. A multidisciplinary research team led an iterative process to develop an overall strategy and specific interventions, based on a theory of change informed by formative research conducted in Atwima Mponua district. The process involved preparatory work, four co-development workshops (August 2021 to November 2022), collaborative working groups to operationalise intervention components, and obtaining ethical approval. Stakeholders including affected individuals, caregivers, other community members and actors from different levels of the health system participated in co-development activities. We consulted these stakeholders at each stage of the research process, including discussion of study findings, development of our theory of change, identifying implementable solutions to identified challenges, and protocol development. Participants determined that the

**Funding:** The Skin Health Africa Research Programme (SHARP) is a collaboration between the London School of Hygiene and Tropical Medicine in the UK, the Noguchi Memorial Institute for Medical Research and the Kumasi Centre for Collaborative Research into Tropical Medicine of Kwame Nkrumah University of Science and Technology in Ghana, and the Armauer Hansen Research Institute and Addis Ababa University in Ethiopia. This project is funded by the National Institute for Health and Care Research (NIHR) under its Research and Innovation for Global Health Transformation (RIGHT) Programme [Grant Reference Number NIHR200125] awarded to ROP, SLW, DYM, MM, CP and RP. The views expressed are those of the author(s) and not necessarily those of the NIHR. The funders had no role in study design, data collection and analysis, decision to publish, or preparation of the manuscript.

**Competing interests:** The authors have declared that no competing interests exist.

intervention should broadly address wounds and other skin conditions, rather than only skin NTDs, and should avoid reliance on non-governmental organisations and research teams to ensure sustainable implementation by district health teams and transferability elsewhere. The overall strategy was designed to focus on a decentralised model of care for skin conditions, while including other interventions to support a self-care delivery pathway, community engagement, and referral. Our theory of change describes the pathways through which these interventions are expected to achieve the strategy's aim, the assumptions, and problems addressed. This complex intervention strategy has been designed to respond to the local context, while maximising transferability to ensure wider relevance. Implementation is expected to begin in 2023.

## Introduction

The World Health Organization (WHO) Neglected Tropical Diseases (NTD) Road Map for 2021–2030 [1, 2] set global targets for the control, elimination, and eradication of 20 diseases. Traditionally, NTD programmes have been predominantly vertical, but the new strategy advocates integration of activities across NTDs and incorporation within the general health system. NTDs involving the skin (skin NTDs) have received less funding and attention than NTDs managed primarily through mass drug administration, and consequently intervention activities have often been sporadic, donor driven, and limited in scope [3]. Integration has emerged as an approach with potential to strengthen activities focused on skin NTDs because of the essential need for complex clinical assessment in diagnosis; their relatively low prevalence and focal nature; and a requirement for prolonged care for the primary infection and associated complications.

Ghana reports the highest number of yaws cases in Africa [4] and the second-highest for Buruli ulcer (BU) [5]. Leprosy is less common (200–300 cases are diagnosed annually), while scabies, lymphatic filariasis, and cutaneous leishmaniasis are endemic. BU and yaws commonly affect children and are predominant in central and southern Ghana, whilst leprosy is reported nationwide. These diseases have a substantial impact on quality of life, resulting in disability, stigma and sometimes mental distress [6], for affected individuals and family members [7–10].

In 2021, Ghana developed an NTD Masterplan, which focused on integrated approaches and multisectoral collaboration [11] but implementation requires collaborative intervention development, adapted to local contexts. Our formative research in Atwima Mponua identified factors shaping care-seeking, diagnosis, and management of skin NTDs and other skin conditions [12]. We found limited community awareness of skin NTDs signs and symptoms, complex moral associations, an absence of well-trained staff, a complex and fragmented healthcare landscape, and the economic burden of care-seeking exacerbated the burden of these diseases. School children in the district with skin NTDs also face substantial social exclusion by peers and staff. While studies have demonstrated that case-finding activities can be integrated across skin NTDs [13–15], there is little evidence on how best to integrate health system responses more broadly to build sustainable holistic models of skin care [16, 17].

Here, we describe the development of an integrated and decentralised skin health strategy to improve experiences of skin NTDs in Atwima Mponua district in Ghana. We outline the processes through which a theory of change, (ToC), informed by formative work, was developed and describe plans for implementation within existing health system structures. The intervention will be evaluated using a before-and-after design. While designed to respond to

the specific context of the study district, this work is also intended to produce generalisable findings regarding the implementation of global and national policies promoting integrated approaches for skin NTDs.

## Methods

### Ethics statement

This project was conducted as part of the Skin Health Africa Research Programme (SHARP) which is a collaboration between the London School of Hygiene & Tropical Medicine (UK), the Kumasi Centre for Collaborative Research into Tropical Medicine and the Noguchi Memorial Institute of Medical Research (both in Ghana), the Armauer Hansen Research Institute and Addis Ababa University (both in Ethiopia). Ethical approval for the formative research and development of the intervention described in this article was obtained from the Noguchi Memorial Institute for Medical Research (NMIMR), Ghana (022/20-21) and the London School of Hygiene and Tropical Medicine, UK (LSHTM Ethics Refs: 22604, 28445). Approval for implementation and evaluation of the proposed intervention has also been obtained from the Ghana Health Services Ethics Committee (GHS-ERC 010/03/23) and the Committee of Human Research, Publication and Ethics (CHRPE), Kwame Nkrumah University of Science & Technology, Kumasi (CHRPE/AP/153/23). All participants provided written informed consent. While different aspects of the intervention development process are presented sequentially, they reflect an iterative process, which began prior to the official start of SHARP in 2019 and will continue through implementation, expected to begin in 2023.

### Study setting

The study district, Atwima Mponua (population: 155,254), was selected because it was considered endemic for several skin NTDs, but no non-governmental organizations (NGOs), civil society organisations (CSOs) or community support groups (CSGs) provide support for affected people. It is one of the largest districts in the Ashanti Region and comprises 7 sub-districts; the district capital is Nyinahin. Forest reserves represent 40% of the total land area, with only limited land clearing for indigenous agriculture. The working population of the district engages in farming, trading and surface mining.

Atwima Mponua has one district hospital and 15 smaller health facilities, including nine health centres and six Community Health Planning and Services (CHPS) compounds. The hospital provides outpatient care, wound dressing and drug administration services to patients, but lacks a dedicated unit to manage skin diseases. The Kumasi Centre for Collaborative Research (KCCR) functions as a referral laboratory in a limited capacity to process samples but currently routinely receives skin NTD samples from only 3 health facilities in the district. Access to health care can be challenging due to the poor road network, limited access to transport, staff shortages, and poverty. Individuals often visit traditional healers for their health needs [12].

### Preparatory work to support intervention development

Prior to developing our interventions, we narratively reviewed literature on intervention strategies for leprosy, BU, and yaws. We identified elements that might be adapted to our current study context and integrated this information alongside data collection conducted as part of our formative research.

We used a co-creation approach to intervention development [18, 19]. We sought input from key stakeholders, including affected individuals, caregivers, local community members

and actors at different levels within the health system. This included involvement in research as informants and consultations at each stage of the research process, including discussion of study findings, development of our ToC, and identifying implementable solutions to identified challenges and protocol development.

## Co-creation workshops

From August 2021 to March 2022, we held three initial intervention development co-creation workshops encompassing stakeholders at the national, regional, district and community levels. The first workshop included community members, affected people, and district healthcare staff, to identify community members' intervention priorities. At this workshop, participants ranked their top three priority interventions for implementation. This was also to ensure the potential strategy would be acceptable to communities. The second and third workshops included Ghana Health Service (GHS) skin NTD programme managers; the regional deputy director for public health; representatives of NGOs supporting programmatic skin NTD activities in nearby districts and healthcare workers from the study district. The second workshop focused predominantly on discussing findings of the formative research [12], while the third was used to refine interventions identified from the earlier workshops for potential implementation. Each meeting was facilitated by the SHARP multidisciplinary research team comprising epidemiologists, clinicians (including dermatologists), health economists, and social scientists.

At each workshop, breakout groups comprising researchers and implementers brainstormed ideas to include within a broad strategy. In the first two workshops, separate groups focused on phases in the patient journey: care-seeking, diagnosis, and treatment. Proposed ideas were fed back to meeting participants to allow refinement. Based on the first two meetings, we developed a conceptual ToC framework, which we used to support identification of key health outcomes and output measures, explore proposed mechanisms, and interrogate assumptions during the third workshop. Stakeholders were asked to rank the potential interventions based on their perceived impact, feasibility, and cost. This ranking was used to inform selection of interventions to include in the final strategy.

## Finalising intervention components

Once key intervention areas had been established, working groups were tasked with developing detailed intervention designs. The community intervention group included social scientists; the supply chain and information systems intervention group included health economists and project managers; and the decentralised service delivery and health worker training groups included clinicians, epidemiologists and social scientists. Over approximately three months, groups met online and by email to describe intervention designs detailing activities involved, material inputs, key stakeholders, and expected outcomes. They articulated assumptions concerning how the intervention would lead to impact, major risks, and potential unintended consequences. Guided by an intervention reporting checklist (S1 Checklist) [20], they described how interventions would be delivered, identified individuals responsible for delivery and supervision, and outlined provisional timelines and budget implications. Finally, they drafted a framework for monitoring and evaluation. Intervention summaries from each group were compiled into a single ToC (Fig 1).

In November 2022, a two-day intervention finalisation workshop was held to discuss the practicality of operationalising intervention activities and further refine implementation plans. Attendees included the deputy director for public health, representatives from the district health management team, healthcare workers from every health facility across the study district, and members of the study team.

| | |
|---|---|
| **Assumptions** | • Pre-existing motivations around care seeking are driven by practical and financial considerations rather than causation beliefs that exclude biomedical framings<br>• Local stakeholders including district- and region-level officers are committed to coordinate and support the interventions<br>• Health workers are receptive, can apply learning and adhere to quality protocols<br>• The government health sector partners will ensure drug and service availability; logistics and supply chains are feasible; health facilities have space, time and resources<br>• Affected individuals with less severe conditions are willing and able to manage their conditions at home, if provided with the means and support to do so<br>• Community members engage with channels used to communicate messaging, and non-GHS actors are open to referring suspect cases |

| **Interventions** | **1. DECENTRALISED CARE MODEL** | **2. SELF-CARE DELIVERY PATHWAY** | **3. COMMUNITY ENGAGEMENT /REFERRAL** |
|---|---|---|---|
| **Activities** | a) Establish and implement wound management protocols at health facility level<br>b) Clinical training package and delivery cascade, provision of job aids and tools<br>c) Provision of diagnostics materials, wound care packs, treatment packs to health centres and CHPS<br>d) Strengthen information systems, including patient tracking forms and monitoring systems for stocks, flows and supply chain<br>e) Establish weekly wound care clinics<br>f) Supportive supervision of facility-based health care workers by district disease control officer | a) Training in patient assessment algorithm to identify clinically suitable patients.<br>b) Provision of condition-specific self-care plan booklets for patients/carers<br>c) Basic management kits to be provided to affected individuals<br>d) Regular review of patients | a) Delivery of mass education events through schools, community durbars and community information centres (including slideshow presentations and storytelling with Q&A and testimonials)<br>b) Engage non-GHS actors (including school health personnel and community health volunteers) to encourage them to refer suspect cases to health facilities |
| **Outputs** | • Affected individuals feel welcome at health facilities, respected, satisfied with their care;<br>• Diagnostic and wound care kits are available at all facilities, and stocks are managed and used appropriately;<br>• Health workers at every facility have the skills to diagnose, treat and manage skin NTDs, wounds and skin problems;<br>• Health workers identify suspected skin NTDs and perform appropriate tests, and results are communicated to patients within set targets of sample being taken;<br>• Appropriate medicines are made available free of charge at the local facility within 7 days of a (lab-confirmed) diagnosis, and all patients complete full course of treatment;<br>• Health centres and CHPS running wound management every two weeks, and receive monthly clinical supervision outreach visits<br>• Patients with more complex conditions receive appropriate timely referral to district-level facility for management | • All patients presenting with wounds meeting defined criteria are offered/provided with appropriate basic management kits and associated counselling;<br>• All individuals following self-care delivery model are monitored regularly at wound clinics | • Increase in NHIS registration and renewal<br>• Increased registration and renewal of the NHIS scheme<br>• Awareness of the availability of skin health NHIS services in the district amongst community members and lay health workers attending outreach events<br>• Acceptance of care for skin problems at health centres and CHPS compounds across various stakeholder groups<br>• Updated evidence-based public health and social communication materials |
| **Intermediate outcomes** | • Improved availability of high-quality diagnosis, treatment and management (including self-care) of skin NTDs and other skin conditions in Atwima Mponua's CHPS and health centres;<br>• Improved acceptability of local GHS services for skin NTDs from both provider and community perspectives;<br>• Improved affordability of local GHS services for skin NTDs to households;<br>• Increased awareness and interest in skin care available locally through the GHS; | | |
| **Distal outcomes** | • Increase in people seeking care for skin problems at GHS facilities within the district;<br>• People presenting symptoms and signs of skin NTDs at GHS facilities receive a timely and accurate diagnosis and/or appropriate referral;<br>• People diagnosed with skin NTDs completing a full package of appropriate treatment; | | |
| **Impact** | Improved wellbeing for people with skin NTDs through reduced physical, psychosocial and economic impact | | |

**Fig 1. Theory of change for the integrated skin health strategy in Atwima Mponua district.**

## Results

In co-development workshops, stakeholders agreed the strategy should aim to improve wellbeing for people with skin NTDs by reducing their negative physical, psychosocial, and economic impact. Participants determined the strategy would need to comprehensively address skin conditions and wounds, rather than skin NTDs in isolation. Despite some concerns that health services could be overwhelmed by the number of individuals with common skin conditions, stakeholders believed that a wider focus on skin health would be more acceptable to people, encourage prompt care-seeking, and might prove an efficient approach to improving health and strengthening health service provision. Participants prioritised developing a strategy that was not reliant on external funding and would in principle be practicable for the district health management team to implement in the study district and, if successful, beyond.

Health system stakeholders ranked improving community awareness of skin lesions and services and training staff as the activities likely to have the highest impact, whilst researchers ranked decentralisation of care and improving community awareness of skin conditions and services as highest impact (Fig 2). Both groups of stakeholders identified community awareness and staff training as most feasible to deliver.

The strategy therefore focuses on an intervention to decentralise care for skin conditions, while including interventions to support a self-care delivery pathway, community engagement, and referral (Fig 1). The following sections describe each of these interventions in turn.

Some interventions were considered but excluded from our final strategy. For example, formal involvement of traditional healers and those selling allopathic medicines (e.g. drug sellers) within the referral system was considered a potential mechanism to improve linkage to care; however, concerns that such a role might be viewed as official endorsement of informal actors with potential negative consequences meant this component was not pursued.

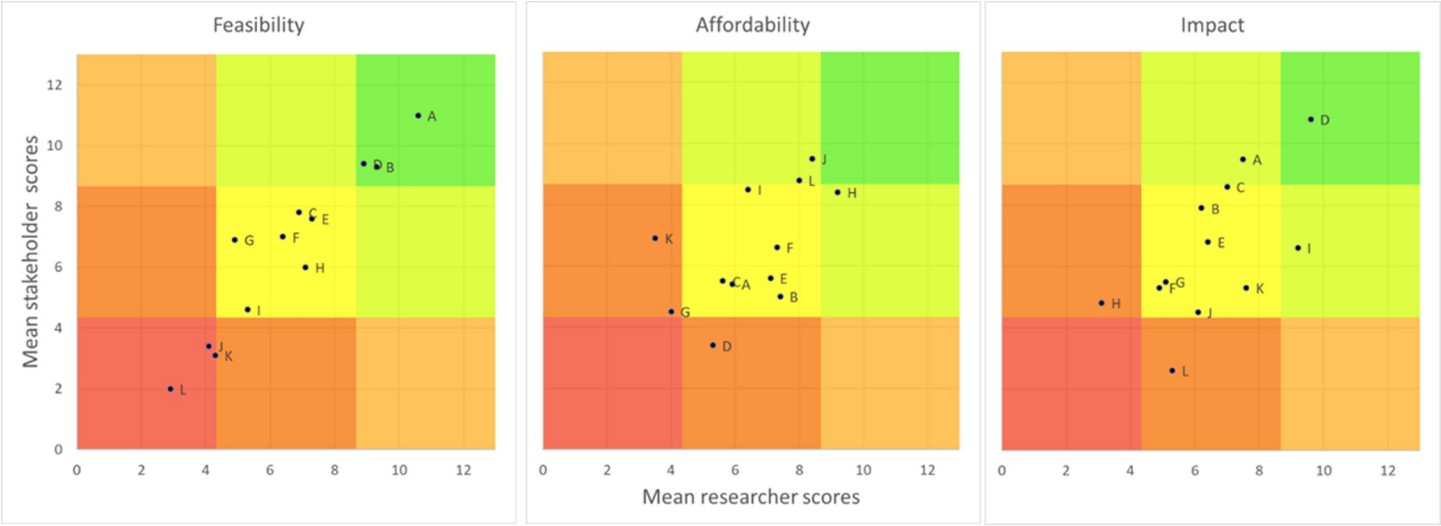

**Fig 2. Prioritisation exercise results.** Participants ranked 12 intervention options across three domains: feasibility, affordability, and potential impact. Options were scored based on their ranking within each domain. Twelve points were assigned for every top ranking, eleven for each second ranking and so on, with one point assigned for the lowest ranking option. Mean scores for each option were calculated for researchers (vertical axis) and health system stakeholders (horizontal) separately. The total number of participants ranking each option highest was calculated and listed in the legend sequentially for feasibility, affordability, and impact. Numbers in brackets show the number of people voting this option highest for (feasibility) (affordability) (impact) of 45 participants (27 researchers and 18 stakeholders), A = Community awareness on skin problems and services through videos and posters to promote care seeking (20) (4) (10), B = Community awareness on skin problems and services through patient testimonials to promote care seeking (2) (3) (3), C = Building a community-of-practice linking traditional healers, volunteers and drug sellers with health facilities and district staff, focused on referral (0) (1) (0), D = Training for health staff on skin diseases, sample collection and management of skin problems (3) (0) (9), E = Training for health staff on respectful care (0) (1) (1), F = Developing a supervisory system for performance monitoring and to support local innovations in health centres and CHPS (2) (3) (0), G = Providing funds to support transport of samples (2) (1) (0), H = Using KCCR as a hub for supplying sample collection consumables (4) (9) (0), I = Decentralising care for skin conditions and skin NTDs to health centres and CHPS (0) (0) (9), J = Dispense drugs to the health centre / CHPS rather than patients collecting from a district/referral hospital (1) (5) (1), K = Availability of a wound care kit at health centres / CHPS free to patients (1) (0) (1), L = Package to support self and family care in the community (0) (8) (1).

## Intervention 1: Train, organise and equip health workers to provide decentralised care for skin conditions

The decentralised care model incorporates patient care pathways and wound management protocols to ensure people affected by common skin conditions, wounds and skin NTDs can: (i) obtain a timely diagnosis, (ii) receive medication and wound care, and (iii) receive guidance to promote wound healing, all from their closest (or preferred) GHS health facility, whilst minimising the cost of care. Pathways (S1 Text) have been established for BU (S1 Table), yaws (S2 Table), leprosy (S3 Table), other wounds (S4 Table), and common skin problems (S5 Table). Protocols are based on WHO and GHS recommendations.

Wound management protocols are largely generic once the underlying aetiology has been addressed and represent a logical platform for integrated service delivery. However, by providing decentralised care, we expect an increased number of individuals with wounds requiring regular review, which may create a strain on health facilities. We will therefore support facilities to establish weekly dedicated wound care clinics. We hypothesise dedicated clinics will support better planning, will represent a visible and accessible point of care, and may facilitate the establishment of informal peer support networks. Together this will support adherence to individualised care plans.

Activities are focused on providing healthcare workers with the training, resources, information systems, support and supervision needed to enable them to successfully implement these protocols in dedicated wound care clinics. All 15 primary health facilities will participate in intervention delivery.

a) **Provide a comprehensive** *training programme for health workers.* Training on skin diseases and intervention activities will be delivered to healthcare workers at each level of the health system. We have developed a curriculum to address learning needs structured to cover four domains of knowledge: i) health system structures, pathways and reporting mechanisms; ii) diagnosis of skin problems; iii) clinical management including wound care and; iv) holistic care encompassing prevention of disability and addressing stigma.

A training of trainers (ToT) approach will be employed to promote sustainability. A separate module will include coaching on subsequent delivery of the training cascade and supportive supervision of junior staff. Master trainers, including senior staff from the district, will be trained by the study teams and will then be responsible for training two health workers per facility, plus sub-district Disease Control Officers (DCOs). Trainees will be responsible for cascading training to other facility and community-based health workers, school health education programme (SHEP) representatives, and community-based health volunteers (CHVs). The training combines classroom teaching, supervised training in clinical examination, and on-the-job supervision. We envisage delivery of an initial district-wide training programme followed by interval refresher training to provide an opportunity both to reinforce learning and to address emergent problems.

b) **Strengthen supportive supervision of facility-based health care workers.** An appropriate supervisory framework embedded within routine management pathways [21] was considered essential. The evidence for supportive supervision interventions is mixed, with some studies showing benefits [22], and others failing to demonstrate service improvement [23]. Reasons for success are complex, with social support, interpersonal interaction, and direct task assistance considered most influential in clinical and other supervisory settings [24, 25]. Interventions were therefore co-developed to ensure alignment with existing organisational infrastructure and operations with a specific focus on task assistance [25]. Task assistance is defined as support provided directly by the supervisor–in this case, DCOs–related to the activity of interest, including direct instruction, advice, and coaching.

We developed project-specific supervision tools and will train district DCOs in their use. Training will be cascaded by DCOs to staff responsible for supervision. Training includes components on the role of supportive supervision, task assistance tools and activities, schedules and reporting, and methods of communication on complex cases and implementation challenges. We aim to leverage existing support supervision frameworks within the district using a dedicated skin care checklist. Data will be reviewed at monthly meetings with the District Director of Health and DCOs. We envisage providing training in supportive supervision, combined with the checklist, will lead to better care provision and improved use of data to review service delivery.

c) **Provision of diagnostic materials, wound care packs and treatment packs to health facilities.** Formative research revealed that few health centres held wound care materials in stock, despite their inclusion in the list of essential medicines available through NHIS [26]; availability of diagnostic materials and medication was also limited; and medicines for skin NTDs were held at regional stores. We will equip facilities with the diagnostic materials, drugs, and wound care materials required to ensure that they are available when individuals seek care (S6 Table). The study team will provide treatment for common skin problems and wound care packs. We will work with district and regional officers to have the anti-microbials used to treat BU and leprosy stored at the district hospital pharmacy (to reduce the time to commencing appropriate antimicrobial therapy) and of dedicated yaws diagnosis and treatment kits (combining rapid diagnostic tests and azithromycin) at facilities.

A hybrid approach, with initial procurement and delivery led by the research team, and ongoing supply managed by the local district health team, is intended to provide proof of

concept for the strengthened supply chain. The study team will procure resources, package ready-to-use kits, and provide an initial supply to health facilities, with the remaining stock held at district stores. The supply chain will be overseen according to routine procedures. DCOs will monitor stock as part of the supportive supervision framework. The district health management team, supported by the study team at KCCR, will be responsible for monitoring essential skin care supplies through a dedicated tracker.

**d) Strengthen information systems (record keeping, reporting and tracking).** To ensure diagnostic and treatment results are recorded and communicated to all levels (including the patient) in a timely manner and to monitor quality of care, information systems require strengthening. We will ensure facilities are stocked with purpose-designed wound care reporting forms developed by the study team, adapted from existing WHO/GHS forms, which capture information on diagnostics, treatment, and outcomes. These forms will be used to document assessments at enrolment and at each follow-up visit. We will also include routine health information tools used for reporting skin NTDs. We will use a secure, mobile phone-based communications platform (WhatsApp) to facilitate requests for tests and medicines between levels of the health system and to alert health facility staff of positive test results reported through the usual GHS system. No patient identifiable data will be shared using this platform. The district DCO, and a representative each from KCCR and NMIMR will administer this group.

The district office will maintain an electronic register in Microsoft Excel of all reported possible cases of BU and leprosy and confirmed cases of yaws. All electronic records will be stored on password protected files. This register will include dates of initial presentation, sample collection, diagnostic testing, and reporting of the result, as well as demographic and clinical data.

## Intervention 2: Supported self-care delivery pathway

Self-care was proposed as a core intervention to reduce the need to travel frequently for treatment and wound dressings, and thereby reduce the costs to patients and caregivers of receiving care and the burden on the health service. It also builds on existing common practices of home-based treatments with plant-based and allopathic medicines in the district which are mainly unsupervised by health facilities [12]. Self-care models have previously been used to prevent leprosy ulcers [27] and in lymphatic filariasis [28]. Self-care includes behaviour undertaken by affected individuals and caregivers to promote or restore health and has proven essential for enhancing wellbeing [29]. The study team will provide the training and materials to GHS health workers in the facilities, who will support self-care within routine services.

After diagnosis, individuals will be invited to a screening assessment for wound self-care. Individuals meeting pre-defined criteria and willing to perform self-care will receive appropriate training from a health worker at their local facility who has been trained and will act as a point of contact. Further, instructions will be provided on returning for regular review or sooner, if needed.

**A) Identification of patients suitable for supported self-care.** The following criteria for wound self-care have been devised:

i. Cutaneous ulcers manageable in community-based settings (<15cm, not involving joints or other important anatomical structures such as eyes or genitalia), or BU category I or II, leprosy-associated ulceration not requiring surgical intervention

ii. An affected adult with a nominated support partner or the primary adult caregiver of an affected child willing to undertake home based wound care

iii. The affected individual has a suitable environment and support from an appropriate carer for wound care at home

iv. Following training, an affected adult or caregiver of an affected child understands the diagnosis and treatment, can apply a wound dressing appropriately and recount the features of adverse changes in the wound such as secondary infection

Individuals not meeting these criteria will be offered care at the health facility according to standard treatment plans.

**B) Provision of self-care booklets & self-care packs.** Individuals undertaking wound self-care will be provided with a self-care plan booklet developed by the study team which contains information about medication, dressing changes, and exercises to prevent contractures. Treatment packs containing sufficient medicines, dressings, and emollients required until the next appointment will be dispensed. The health worker will discuss and agree with the patient and support partner a care plan to promote adherence to medication and dressings changes at home. Affected individuals needing additional support will be invited to attend more frequently.

**C) Clinical review.** Individuals and their wounds will be assessed at weekly wound care clinics which will provide further opportunity for health workers to address issues associated with self-care, emphasize educational messages and explore any concerns. Wounds not healing satisfactorily or deteriorating will be an indication for a switch to facility-based care. Any change in an individual's circumstances which may prevent them from undertaking self-care may also be deemed an indication for facility-based care.

### Intervention 3: Community engagement and referral

The historic lack of local GHS services for people with skin disease in Atwima Mponua was identified as one of the reasons individuals often used home remedies and sought care from traditional healers and drug vendors. To encourage care-seeking when decentralised care becomes available and reduce social exclusion, we will develop communications materials and work with health promotion staff (community health volunteers (CHVs), School Health Education Programme (SHEP) coordinators and facility-based staff) to use existing GHS health promotion channels to implement a community engagement campaign. This campaign will seek to: (i) inform communities about improved services for skin health, (ii) encourage those with cutaneous symptoms to seek care from GHS facilities, (iii) promote enrolment and/or renewal of NHIS membership to address economic barriers to treatment-seeking, (iv) involve informal health sector actors in community events, (v) address school information needs about Infection Prevention and Control (IPC) and (v) collect and address feedback on intervention implementation. CHVs and SHEPs will also be engaged to encourage referrals to decentralised services.

**A) Skin health promotion events.** Skin health promotion activities will be delivered by health promotion staff trained using the GHS ToT model as part of established modes of delivery to target populations at key locations. Messaging will be developed in consultation with community members and stakeholders and tailored to each type of event. Health promoters will be trained in feedback reporting by the research team. The modes of delivery are:

Community durbars: In regular community durbars (meetings), which GHS already uses, health promoters will provide messaging about skin diseases and services in the form of multimedia presentations, testimonials, and role plays developed by the study team. We will explore the feasibility of training former patients, health personnel and traditional healers to share testimonies and act as champions for decentralised skin care services within these CHV-led

activities. Staff of drug shops will be invited to attend and participate. CHVs will run dynamic question and answer periods to encourage participation and learning and ask attendees to share their experiences of skin care services. Key questions, concerns and themes from discussion periods will routinely be noted by health promotion staff and communicated to supervisors in reports and monthly supervision meetings, where opportunities for further action, programme refinement and feedback to target populations will be discussed.

School-based education campaigns: We will train SHEP coordinators and CHVs from each of the health facilities to deliver key messages in their monthly health promotion campaigns at schools. Activities will be child-friendly, involving fun facts about the skin and focusing on how to be a good friend when others experience skin diseases. Teachers and other school staff will be engaged in discussions to answer their questions about managing children with skin conditions at school, reflect on IPC practices which can inadvertently stigmatise, and share information about services with parents of children with skin conditions. SHEP coordinators will report on key questions and themes to supervisors for further action and feedback, as above.

Community information centres (CICs): Typically operated from small buildings, with simple equipment including microphones and amplifiers, CICs are a common form of information dissemination. We will train health promoters to share simple messages promoting the new skin health services.

Film-based activities: GHS organises regular film screenings in communities and health promoters provide additional health messaging and news during intermissions. We will work with GHS to communicate relevant messages about skin services during news sessions, including specific information on how communities can access their nearest service providers.

**B) Community-based referrals.** In addition to training CHVs and SHEP coordinators to deliver the above skin health promotion activities, we will ask them to opportunistically champion, share information and answer questions about skin health services when they encounter people with skin diseases in their other daily work. For SHEP coordinators, this includes monthly activities at schools on rotating health themes.

## Implementation

Implementation is expected to last twelve months. To support sustainability, the district health management team will implement major activities. The study team will train and mentor the district team, guide prioritisation of activities, prepare and facilitate cascade training and community engagement activities, assemble wound care and medicine packs and provide them to district medical stores, and ensure continuous high-level clinical and logistical support to the DCOs. The study team will not, however, be responsible for day-to-day management and provision of care within facilities or health promotion work in regular education campaigns.

## Discussion

We developed an integrated skin health strategy adapted to the context of Atwima Mponua district using a multi-stage, iterative, consultative process. The strategy aims to improve well-being for people with skin NTDs via a focus on decentralisation of diagnosis and treatment of skin conditions to peripheral health facilities, with components to support self-care and community engagement and referral. Our intervention is aligned closely with the GHS NTD Masterplan [11] and the WHO Roadmap [1, 2]. Delivering the intervention primarily through existing health system structures has the potential to ensure long-term sustainability.

Our strategy differs in important ways from the approach we anticipated prior to conducting formative research. Our original focus on "severe, stigmatising skin diseases," as designated by our project's funding, shifted to "skin NTDs" in line with the evolving policy context and to

avoid stigmatisation. Reflective practice and engagement with stakeholders broadened our scope further from skin NTDs to skin health as concerns were raised about the effectiveness and cost-effectiveness of a strategy that would not include treatment for most people with skin problems. We originally conceived an "integrated case finding and management" strategy [30], but our formative research revealed limited access to timely, high-quality, affordable care–rather than "finding" cases–was the significant limiting factor to improving outcomes, leading to our focus on decentralised care. We originally envisaged supporting a broad community of practice including traditional healers, as had been done successfully in other countries [31]; however, stakeholders did not support this approach in Ghana. Our findings thus demonstrate the importance of formative research and iterative engagement with a broad range of stakeholders in co-creating an acceptable implementation plan for an "integrated skin NTD strategy".

Our developed strategy builds on the existing literature in important ways. Several studies have attempted to address barriers to care for one or more skin NTDs both in Ghana and in other contexts [32–35]. Active case-finding activities have previously been shown to identify more cases than would otherwise present to the health service [34, 36], but these approaches neither tackle fundamental barriers to care-seeking, nor result in widespread improvements in the care available, and have required substantial financial support and delivery by a dedicated research or implementation team. There is evidence to suggest decentralised care models for BU appear to be both effective and sustainable [33]. Decentralisation aligns with the Ghana NTD Masterplan [11], the NTD roadmap agenda for incorporating NTD care into general services, and Universal Health Coverage strategies [1].

While some aspects of our intervention strategy may be specific to Atwima Mponua, other aspects and the methods we used may be more generalisable. An integrated NTD strategy is necessarily a complex intervention [37], not just because the individual diseases are clinically complex [1], but because of the nature of the interventions, their interdependence, and their interactions with context. We describe in this article (and in a report of our formative research) the steps we employed to understand the local context and engage meaningfully with stakeholders to develop this strategy using an explicit ToC. The activities within our strategy may need to be adapted for other districts in Ghana or in other countries, with the degree of adaptation dependent on the similarity of the target context to Atwima Mponua [38, 39]. Nonetheless, we would expect the mechanisms of action through which activities are expected to lead to overall impact could be more widely relevant [38, 39]. In particular, the problem of limited access to care for skin NTDs is widespread, which creates potential for our strategy, if shown to be effective and cost-effective, to be transferable. To enhance transferability, our proposed strategy will be delivered almost exclusively by the GHS predominantly using existing structures.

Our intervention development approach has strengths and limitations. Firstly, we used the TIDieR checklist to ensure all intervention elements have been defined. Secondly, it was informed by a large body of formative research [12]. We identified factors shaping wellbeing for people with skin NTDs and examined evidence of what strategies had been successful. Thirdly, the intervention was developed over a series of events and ongoing engagement with a range of stakeholders, allowing iterative refinement of its scope and focus. One limitation is that our formative work focused predominantly on NTDs, but our intervention will have a broader scope. It is plausible we missed important insights into wider issues of skin care, which may limit the effectiveness and cost-effectiveness of our strategy. Although we held multiple meetings, not all stakeholder groups were represented at every workshop, and it is possible our final intervention does not fully capture the concerns of all relevant groups. Finally, we

opted not to include some potential components such as direct involvement of traditional healers. Future studies addressing the added value of these components would be valuable.

The strategy we developed has several implications for NTD programmes and research. WHO advocates for mainstreaming of activities within the health service and integration of interventions across NTDs; however, we have shown that addressing the burden of skin NTDs also requires integration across the care pathway and strengthening of wider health system functions. This requires considering the capacities and social dynamics of primary healthcare facilities to engage with skin NTD care, alongside other spaces at the margins of the health system such as schools, pharmacies and sites of traditional and religious healing [27, 40]. It is important for policymakers, programme managers, and academics to recognise control of skin NTDs requires complex interventions in complex systems [37, 41–43]. While vertical MDA programmes may successfully tackle some NTDs, replicating this approach–for example, with strategies focused only on case detection integrated across diseases–will not address the key problems facing people with skin NTDs in many contexts. Alongside the intervention implementation, we will conduct a rigorous, multidisciplinary evaluation to assess the effectiveness, costs, and cost-effectiveness of our strategy and the processes by which the planned activities contribute–or not–to intended and unintended outcomes.

## Supporting information

**S1 Text. Patient care pathways, wound management protocols and wound-care clinics.**
(DOCX)

**S1 Table. Patient care pathways for BU.**
(DOCX)

**S2 Table. Patient care pathways for Yaws.**
(DOCX)

**S3 Table. Patient care pathways for Leprosy.**
(DOCX)

**S4 Table. Patient care pathways for Wounds and non-BU ulcers.**
(DOCX)

**S5 Table. Patient care pathways for common skin problems (Scabies, impetigo, Tinea capitis, Tinea corporis).**
(DOCX)

**S6 Table. Diagnostics, wound care and treatment packs provided to facilities through the interventions.**
(DOCX)

**S1 Checklist. The TIDieR (Template for Intervention Description and Replication) checklist\*: Information to include when describing an intervention and the location of the information.**
(DOCX)

## Acknowledgments

We wish to thank the individuals and communities for their participation in the work of the Skin Health Africa Research Programme. We sincerely acknowledge staff of the Atwima

Mponua District Health Directorate. We wish to thank the policy makers within the Ghana Health Service who dedicated their time to participate in the development of this intervention.

**Membership of SHARP collaboration**

Endalamaw Gadisa (Armauer Hansen Research Institute, Ethiopia), Mirgissa Kaba (Addis Ababa University, Olivia Dornu (Kumasi Centre for Collaborative Research), Saba Lambert, Sinead Langan (London School of Hygiene and Tropical Medicine, UK)

## Author Contributions

**Conceptualization:** Richard Odame Phillips, Eric Koka, Jacob Novignon, Jennifer Palmer, Collins Ahorlu, Yaw Ampem Amoako, Stephen L. Walker, Dorothy Yeboah-Manu, Michael Marks, Catherine Pitt, Rachel Pullan.

**Data curation:** Lucy Owusu, Hope Simpson, Abigail Agbanyo, Daniel Okyere, Michael Ntiamoah Oppong, Iris Mosweu.

**Formal analysis:** Lucy Owusu, Edmond Kwaku Ocloo, Hope Simpson, Jacob Novignon, Tara B. Mtuy, Jennifer Palmer, Yaw Ampem Amoako.

**Funding acquisition:** Richard Odame Phillips, Stephen L. Walker, Dorothy Yeboah-Manu, Michael Marks, Catherine Pitt, Rachel Pullan.

**Investigation:** Lucy Owusu, Edmond Kwaku Ocloo, Abigail Agbanyo, Daniel Okyere, Ruth Dede Tuwor, Yaw Ampem Amoako.

**Methodology:** Richard Odame Phillips, Eric Koka, Jacob Novignon, Tara B. Mtuy, Jennifer Palmer, Collins Ahorlu, Yaw Ampem Amoako, Stephen L. Walker, Dorothy Yeboah-Manu, Michael Marks, Catherine Pitt, Rachel Pullan.

**Project administration:** Richard Odame Phillips, Ruth Dede Tuwor, Adelaide Fokuoh-Boadu, Richard Adjei Akuffo, Adwoa Asante-Poku, Yaw Ampem Amoako.

**Resources:** Jojo Cobbinah.

**Supervision:** Tara B. Mtuy, Collins Ahorlu, Michael Marks, Rachel Pullan.

**Validation:** Abigail Agbanyo, Ruth Dede Tuwor, Jacob Novignon, Michael Ntiamoah Oppong, Iris Mosweu, Adwoa Asante-Poku, Jojo Cobbinah, Stephen L. Walker.

**Visualization:** Daniel Okyere, Adelaide Fokuoh-Boadu, Richard Adjei Akuffo, Michael Ntiamoah Oppong, Iris Mosweu, Adwoa Asante-Poku, Jojo Cobbinah.

**Writing – original draft:** Richard Odame Phillips, Lucy Owusu, Eric Koka, Tara B. Mtuy, Jennifer Palmer, Collins Ahorlu, Yaw Ampem Amoako, Michael Marks, Catherine Pitt, Rachel Pullan.

**Writing – review & editing:** Richard Odame Phillips, Lucy Owusu, Eric Koka, Edmond Kwaku Ocloo, Hope Simpson, Abigail Agbanyo, Daniel Okyere, Ruth Dede Tuwor, Adelaide Fokuoh-Boadu, Richard Adjei Akuffo, Jacob Novignon, Michael Ntiamoah Oppong, Iris Mosweu, Adwoa Asante-Poku, Jojo Cobbinah, Tara B. Mtuy, Jennifer Palmer, Yaw Ampem Amoako, Stephen L. Walker, Dorothy Yeboah-Manu, Michael Marks, Catherine Pitt, Rachel Pullan.

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
