## [Decision Letter · Decision Letter 0]

26 Sep 2023

PGPH-D-23-01377

Development of an integrated and decentralised skin health strategy to improve experiences of skin neglected tropical diseases in Atwima Mponua District, Ghana

Dear Dr. Phillips,

Thank you for submitting your manuscript to PLOS Global Public Health. After careful consideration, we feel that it has merit but does not fully meet PLOS Global Public Health’s publication criteria as it currently stands. Therefore, we invite you to submit a revised version of the manuscript that addresses the points raised during the review process.

We look forward to receiving your revised manuscript.

Kind regards,

Jianhong Zhou

Staff Editor

Journal Requirements:

2. Please provide separate figure files in .tif or .eps format only and remove any figures embedded in your manuscript file. Please also ensure all files are under our size limit of 10MB.

Additional Editor Comments (if provided):

Reviewers' comments:

Reviewer's Responses to Questions

**Comments to the Author**

1. Does this manuscript meet PLOS Global Public Health’s publication criteria? Is the manuscript technically sound, and do the data support the conclusions? The manuscript must describe methodologically and ethically rigorous research with conclusions that are appropriately drawn based on the data presented.

Reviewer #1: Yes

Reviewer #2: Yes

2. Has the statistical analysis been performed appropriately and rigorously?

Reviewer #1: N/A

Reviewer #2: Yes

3. Have the authors made all data underlying the findings in their manuscript fully available (please refer to the Data Availability Statement at the start of the manuscript PDF file)?

Reviewer #1: Yes

Reviewer #2: Yes

4. Is the manuscript presented in an intelligible fashion and written in standard English?

Reviewer #1: Yes

Reviewer #2: Yes

5. Review Comments to the Author

Reviewer #1: This is an excellent paper in preparation for its full implementation.

Few comments

1. Reference to WHO Framework of skin NTDs is missing (https://www.who.int/publications/i/item/9789240051423). It is a companion document to the NTDs road map.

2. Is there a role of private healthcare providers and facilities? I only see reference to GHS facilities.

3. Page 7 last line: Can you please add reference to the essential medicines list from the NHIS?

4. Page 8 (d): Use of Whatsapp is a good and effective medium to communicate. Confidentiality is sometimes raised as an issues. What steps will be taken to protect participants confidentiality?

5. Page 8 (intervention 2). Burden on families should also be mentioned especially caregivers/parents of children. Not only on the health service.

6. Page 9: For self-care of wounds, i would suggest <10 cm and not located at a site where restrictions of movement (joint or otherwise) can occur or damage to important organs.

7. Page 9 (clinical review): I will suggest ...emphasize educational messages rather than promote.

8. Page 9 (clinical review): I will suggest ....wounds not healing satisfactorily instead of not progressing satisfactorily.

9. Additional reference that may be useful: https://www.ncbi.nlm.nih.gov/books/NBK553823/

Reviewer #2: PLoS PGPH-D-23-01377 REVIEW

REPORT

1. Summary

A comprehensive and exhaustive study by a multidisciplinary research team aimed at developing an overall strategy and specific interventions based on a theory of change informed by formative research works previously conducted in Atwima Mponua district Ghana.

The strategy used focuses on 3 prongs: an intervention to decentralise care for skin conditions, interventions to support a self-care delivery pathway with community engagement, and referral of cases.

It is a sustainable programme that can help to achieve the WHO skin NTDs goals as well as management of prevailing skin diseases using existing health structures and community health programmes .

2. Minor corrections are required.

Title and Abstract

Paragraph 3

As strategies are for all skin diseases and not only NTDs, the title should reflect this so the study can be seen by others to be for other skin conditions other than skin NTDs. For example: “Development of an integrated and decentralised skin health strategy to improve experiences of skin neglected tropical diseases and other skin conditions in Atwima Mponua District, Ghana”

Introduction

The introduction was well written and in the context of the title and aim of the study. Based on World Health Organization (WHO) Neglected Tropical Diseases (NTD) Road Map for 2021–2030, global targets for the control, elimination, and eradication of 20 diseases requires development and integration of strategies or intervention that works for each country, region and district.

State some interventions (or holistic models) that have been made in similar districts within the country or other countries involving NTDs that can be a framework for a similar work by your team.

Stigma is a major issue in the management of skin NTDs. Clearly state steps that will be used to reduce this rather than stating education of the populace so others can use similar interventions.

Methods

Ethical approval was obtained and written informed consent.

Grants or sponsorship or collaborations should be expressly stated here. How is the London School of Hygiene and Tropical Medicine involved? What does this involve - Skin Health Africa Research Programme (SHARP) project in 2019? It is not clear to a reader of this study who does not know your previous works.

The study location was appropriate for this type of study and programmes to be implemented. There will be an immense benefit of this work to the local community and the developing countries.

Study design

This was appropriate and comprehensive. The need assessment through the co-creation approach would ensure easy integration into the health system and implementation of the needed interventions.

The aim for implementable solutions to identified challenges and protocol development would ensure that this is not a work that will remain on paper but implemented by the appropriate authorities .

Does the SHARP multidisciplinary research team include a dermatologist? This should be stated if it does. If it does not, I would like to suggest that this be included in the implementation section. The involvement of a dermatologist will really be useful in training of health personnel as this work is based on a framework of providing care not just for NTDs but all other skin conditions. The WHO recognises that 10 out of the 20 NTDs have skin manifestations. This is buttressed by Figure 2 point E which was considered not feasible but training of health officials is central to addressing WHO NTDs goals.

The development of a conceptual ToC framework, and stakeholders ranking of the potential interventions based on their perceived impact, feasibility, and cost are two strategies worth emulating by others in developing their own interventions.

The use of TIDieR (Template for Intervention Description and Replication) Checklist would ensure reproducibility and adaptability as the authors aimed for in any locality.

Results

The ToT programme is central to implementation and sustainability of health ventures and eradication of diseases.

The delivery of an initial district-wide training programme followed by interval refresher training in order to provide opportunities for reinforcing learning and address identifiable problems is a great strategy.

The plan for sustainability using the district health management team to implement major activities will provide continuity of the program in a forseeable future.

Correct highlighted area in Figure 2 Prioritisation exercise results. There were 45 participants and not 35.

Collaboration and funding statement should be as clear and detailed as in the response on the manuscript draft.

Is there any plan for the National Institute for Health and Care Research (NIHR) to fund a reassessment of this work to assess the impact or evaluate the sustainability of the program in the future such as a 2 or 5 year plan? This can be added if there is a possibility. Sustainability of health programmes is paramount to obtaining NTDs goals.

References

Correct citations. PLOS uses the reference style outlined by the International Committee of Medical Journal Editors (ICMJE), also referred to as the “Vancouver” style. Many journal titles were not abbreviated. For instance, reference 35 page 20 - “International journal of public health” should be “Int J Public Health” Reference 17 should be “BMJ” not “Bmj” as written. “PLoS Negl Trop Dis.” should replace “PLoS neglected tropical diseases” in references 31 and 32.

2

Data is easily accessible. No conflict of interest declared by authors.

6. PLOS authors have the option to publish the peer review history of their article (what does this mean?). If published, this will include your full peer review and any attached files.

**Do you want your identity to be public for this peer review?** For information about this choice, including consent withdrawal, please see our Privacy Policy.

Reviewer #1: No

Reviewer #2: **Yes: **Olumayowa Abimbola Oninla

---

## [Decision Letter · Decision Letter 1]

22 Nov 2023

PGPH-D-23-01377R1

Development of an integrated and decentralised skin health strategy to improve experiences of skin neglected tropical diseases and other skin conditions in Atwima Mponua District, Ghana

Dear Dr. Phillips,

Thank you for submitting your manuscript to PLOS Global Public Health. After careful consideration, we feel that it has merit but does not fully meet PLOS Global Public Health’s publication criteria as it currently stands. Therefore, we invite you to submit a revised version of the manuscript that addresses the points raised during the review process.

We look forward to receiving your revised manuscript.

Kind regards,

Sakib Burza, MBChB, MRCGP, MSc, PhD

Academic Editor

Journal Requirements:

Additional Editor Comments (if provided):

Many thanks for submitting the revised article. It is fit for publication and will be accepted, but we ask that you address the issue with the references first please.

Reviewers' comments:

Reviewer's Responses to Questions

**Comments to the Author**

1. If the authors have adequately addressed your comments raised in a previous round of review and you feel that this manuscript is now acceptable for publication, you may indicate that here to bypass the “Comments to the Author” section, enter your conflict of interest statement in the “Confidential to Editor” section, and submit your "Accept" recommendation.

Reviewer #2: (No Response)

2. Does this manuscript meet PLOS Global Public Health’s publication criteria? Is the manuscript technically sound, and do the data support the conclusions? The manuscript must describe methodologically and ethically rigorous research with conclusions that are appropriately drawn based on the data presented.

Reviewer #2: Yes

3. Has the statistical analysis been performed appropriately and rigorously?

Reviewer #2: Yes

4. Have the authors made all data underlying the findings in their manuscript fully available (please refer to the Data Availability Statement at the start of the manuscript PDF file)?

Reviewer #2: Yes

5. Is the manuscript presented in an intelligible fashion and written in standard English?

Reviewer #2: Yes

6. Review Comments to the Author

Reviewer #2: Review of manuscript PGPH-D-23-01377R1

Development of an integrated and decentralised skin health strategy to improve

experiences of skin neglected tropical diseases and other skin conditions in Atwima

Mponua District, Ghana

The authors have done a comprehensive study. The corrections of the manuscript

has been diligently done save for a few references.

I have checked the each reference. It was noted that the journal names of some

references were not abbreviated.

A quick way: Copy the journal name in the references to Google search bar and add

the word ‘abbreviation’, this will bring out the journal abbreviation. For example, for

reference number 42, ‘Annual review of public health abbreviation’ will be Annu

Rev Public Health.

Reference 9: Correction should be - Trop Med Health.

Reference 10: Correction should be - Trop Med Int Health.

Other references for correction of journal abbreviations to be done are

reference number 20, 24, 31 (use number 40 journal abbreviation), 33, 34,

35, 36, 38, 39, 40, 41 (see below), 42.

For reference 30, add the date cited and the web link:

[Cited _____ (input the date you used it here)] Available from: Developing an

integrated intervention manual for skin NTDs.pdf (lstmed.ac.uk).

Reference 41 - Greenhalgh T, Papoutsi C. Studying complexity in health

services research: desperately seeking an overdue paradigm shift. BMC Med.

2018;16(1):95. doi:10.1186/s12916-018-1089-4

The manuscript is acceptable for publishing after these corrections and may not

need further review work.

7. PLOS authors have the option to publish the peer review history of their article (what does this mean?). If published, this will include your full peer review and any attached files.

**Do you want your identity to be public for this peer review?** For information about this choice, including consent withdrawal, please see our Privacy Policy.

Reviewer #2: **Yes: **Olumayowa Abimbola Oninla

---

## [Editor Report · Decision Letter 2]

19 Dec 2023

Development of an integrated and decentralised skin health strategy to improve experiences of skin neglected tropical diseases and other skin conditions in Atwima Mponua District, Ghana

PGPH-D-23-01377R2

Dear Dr. Phillips,

We are pleased to inform you that your manuscript 'Development of an integrated and decentralised skin health strategy to improve experiences of skin neglected tropical diseases and other skin conditions in Atwima Mponua District, Ghana' has been provisionally accepted for publication in PLOS Global Public Health.

Best regards,

Sakib Burza, MBChB, MRCGP, MSc, PhD

Academic Editor